# Low threshold lasing emissions from a single upconversion nanocrystal

Yunfei Shang [1,2,3], Jiajia Zhou [1✉], Yangjian Cai[3], Fan Wang [1], Angel Fernandez-Bravo[4], Chunhui Yang[2], Lei Jiang [5] & Dayong Jin [1,3]

Cross-relaxation among neighboring emitters normally causes self-quenching and limits the brightness of luminescence. However, in nanomaterials, cross-relaxation could be well-controlled and employed for increasing the luminescence efficiency at specific wavelengths. Here we report that cross-relaxation can modulate both the brightness of single upconversion nanoparticles and the threshold to reach population inversion, and both are critical factors in producing the ultra-low threshold lasing emissions in a micro cavity laser. By homogenously coating a 5-µm cavity with a single layer of nanoparticles, we demonstrate that doping $Tm^{3+}$ ions at 2% can facilitate the electron accumulation at the intermediate state of $^3H_4$ level and efficiently decrease the lasing threshold by more than one order of magnitude. As a result, we demonstrate up-converted lasing emissions with an ultralow threshold of continuous-wave excitation of ~150 W/cm$^2$ achieved at room temperature. A single nanoparticle can lase with a full width at half-maximum as narrow as ~0.45 nm.

[1] Institute for Biomedical Materials and Devices (IBMD), Faculty of Science, University of Technology Sydney, Sydney, NSW 2007, Australia. [2] MIIT Key Laboratory of Critical Materials Technology for New Energy Conversion and Storage, School of Chemistry and Chemical Engineering, Harbin Institute of Technology, Harbin 150001, P.R. China. [3] UTS-SUStech Joint Research Centre for Biomedical Materials and Devices, Department of Biomedical Engineering, Southern University of Science and Technology, Shenzhen, Guangdong, P.R. China. [4] Department of Physics, Politecnico di Milano, Piazza L. Da Vinci 32, Milano 20133, Italy. [5] Laboratory of Bio inspired Smart Interface Science, Technical Institute of Physics and Chemistry, Chinese Academy of Sciences, Beijing 100190, P.R. China. ✉email: jiajia.zhou@uts.edu.au

Cross-relaxation (CR) is an energy transfer process between a pair of nearby emitters, where one at a higher excited energy state transfers its photon energy to the other one at a lower excited state or ground state, so that both can simultaneously reach their intermediate excited states[1]. This process can be often observed in lanthanides ions those featured with a series of sophisticated intermediate energy levels and each with long lifetimes[2,3]. CR accumulates energy in the intermediate excited states, as a conducive process to the formation of population inversion that is essential for the generation of lasing emissions. This explains why lanthanide ions doped laser crystals and glass are ascendant gain medium[4–6]. Though the concentration of lanthanide ions determines the extent of CR and thereby the threshold for lasing emission generation, in the bulk materials the doping uniformity and dynamic range of lanthanide ions are hard to control, e.g., typically 0.25–1% for $Tm^{3+}$ in $YVO_4$, $MgWO_4$ or YAG laser crystal ($<8 \times 10^{20}$ ions/cm$^3$)[7–9]. High doping in bulk crystal and glass often leads to the non-uniform distribution of the ions and localized excessive CR, which results in luminescence quenching, self-heating, and high laser threshold.

At the nanoscale, lanthanides doped upconversion nanoparticles (UCNPs) and wet chemistry synthesis strategies have been well developed, so that their size, shape and doping concentrations can be precisely controlled with high accuracies in both morphological and optical uniformities[10–15]. In a typical $NaYF_4$ host, the doping percentage of a lanthanide ion can be arbitrarily tuned between zero and its unity to form the "alloyed" nanocrystals[16,17]. This allows several recent studies on the role of doping concentrations and the degree of CR between ions in increasing the efficiency of optical depletion in super-resolution microscopy and facilitating the generation of near-infrared and single band emissions[2,18–22]. Encouragingly, singly $Tm^{3+}$ doped UCNPs with energy looping effect have been recently reported as an efficient gain medium to generate room-temperature continuous-wave (CW) pumped laser with a threshold of ~14 kW/cm$^2$ [23].

Here we study the role of CR (doping concentration), excitation power density and the size of a single nanoparticle, and to identify the optimum conditions and key characteristics to set a single nanoparticle to lase. We find that the brightness of UCNPs and its efficiency in establishing the population inversion as the gain medium as well as the quality factors of a typical whispering-gallery-mode (WGM) cavity can be strongly affected by the doping concentration, excitation power density and the size of UCNPs when coating a typical polystyrene microsphere by a single layer of self-assembled UCNPs or only a single UCNP. In the classical $Yb^{3+}$-$Tm^{3+}$ co-doped energy transfer system, we find that the higher doping concentration of $Tm^{3+}$ facilities higher probability for CR, but requires higher laser excitation intensity to produce sufficiently bright upconversion emissions. Moreover, though the larger size of UCNPs typically produces stronger emissions, it generates a stronger scattering loss by the increased roughness of the cavity surface. Since the scattering scales with the 6th power of the particles' diameter, reducing the number of particles will significantly reduce the loses of the cavities[24,25]. This effect can significantly reduce the efficiency for micro cavity lasers coated by a single layer of self-assembled UCNPs, but has negligible influence on the lasing emissions from a single UCNP.

## Results and discussion

**Population inversion property of CR nanoparticles.** TEM images displayed the Fig. 1a of a series of morphology-uniform UCNPs at different $Tm^{3+}$ concentrations show the monodispersity of our nanocrystals. Within the volume of ~7240 nm$^3$, in a 24 nm UCNP (Supplementary Fig. 1), we tune the amount of

the $Tm^{3+}$ ions from ~200 to ~8000, 0.2 mol.% to 8 mol.% correspondingly. As shown in Fig. 1b, when the concentration of $Tm^{3+}$ is fixed at low level, the population inversion purely depends on the high power density of the pumping laser, as the activation of the emitters is mainly through the sensitization of $Yb^{3+}$ (Supplementary Fig. 2) and there is negligible CR due to the long distance between the lanthanide ions, e.g., the averaged large distance can be calculated as ~3.3 nm at the doping concentration of 0.2 mol.%. The dependence on the high power excitation can be alleviated by CRs, ($^1G_4$, $^3H_6 \rightarrow ^3H_5$, $^3H_4$), ($^1G_4$, $^3H_6 \rightarrow ^3F_{2,3}$, $^3F_4$), ($^3F_{2,3}$, $^3F_4 \rightarrow ^3H_4$, $^3H_5$)[18,26], which facilitates the establishment of population inversion at intermediate levels, such as $^3H_4$. Besides, highly $Tm^{3+}$ doped UCNPs leads to the decrease in the $Yb^{3+}$-$Tm^{3+}$ distance and increase in the $Yb^{3+}$-$Tm^{3+}$ energy transfer efficiency. But this strategy, as illustrated in Fig. 1d, could lead to a quenching of the overall upconversion emissions, when doping too many emitters. As shown in Supplementary Fig. 4, the decrease in the $Yb^{3+}$ lifetime values and the trend of energy transfer efficiency saturation appeared at the very large $Tm^{3+}$ concentrations (e.g., >8 mol.%), indicate the possible $Tm^{3+}$-$Yb^{3+}$ energy back transfer[27]. To alleviate the concentration quenching and energy back transfer effects in the highly doped UCNPs, high excitation power density is required to pump the significant amount of the ground level $Tm^{3+}$ emitters and $Yb^{3+}$ sensitizers. These suggest the existence of a sweet spot of optimum $Tm^{3+}$ doping concentration and excitation pumping power for the low threshold establishment of population inversion.

We first characterize the upconversion emission spectra of the series of 24 nm UCNPs (Fig. 1a). Diversified spectral distributions are observed when varying the $Tm^{3+}$ concentrations (Fig. 2a), suggesting the concentration-dependent CR effect. The proportions of the emissions from $^3H_4$ to $^3H_6$ transition show an upward tendency with the increase of $Tm^{3+}$ concentration, which indicates the CR induced efficient energy accumulation (Supplementary Fig. 6). At the low concentration range, the energy, sensitized by and transferred from $Yb^{3+}$, is distributed onto all the excited states, including $^1D_2$, $^1G_4$, and $^3H_4$ (Supplementary Fig. 7). When the $Tm^{3+}$ concentration increases from 1 mol.% to 2 mol.%, the peak intensities associated with the $^1G_4$ level decrease, while the $^3H_4$ initiated transition intensity increases. Figure 2b displays the quantitative intensity evolution and ratios as the function of $Tm^{3+}$ concentration. The intensity at 802 nm ($^3H_4 \rightarrow ^3H_6$) transition, the intensity ratios of 802 nm/473 nm and 802 nm/645 nm reach their peak values at 2 mol.% $Tm^{3+}$ due to the CR induced population enrichment at the excitation power density of 100 W/cm$^2$. Further increase in the $Tm^{3+}$ concentration, e.g. 4 mol% and 8 mol.%, depopulates the $^3H_4$ level with decreased intensity at 802 nm due to the excessive CR induced energy loss and the possible back energy transfer from $Tm^{3+}$ to $Yb^{3+}$. As CR involves different energy levels between a pair of nearby emitters, the population distribution of each energy level is strongly dependent on the excitation power density, so is the CR dynamics[2,28]. And Fig. 2c shows the power-dependent population of $^3H_4$ level by analyzing the peak intensity at 802 nm. The 2 mol.% $Tm^{3+}$ doped sample shows the highest intensities of 802 nm emissions within the excitation power density range of 0.6–150 W/cm$^2$ due to the CR effect. Above the power density of 10 W/cm$^2$, the low doping samples are saturated while the highly doped samples exhibit an accelerated growth trend of the emission intensities, indicating the non-linear transition dynamics caused by the power-dependent and concentration-dependent CR effect.

**Characterization of microlasers coated with UCNPs.** Figure 2d, e and Supplementary Fig. 8 present the typical images from SEM

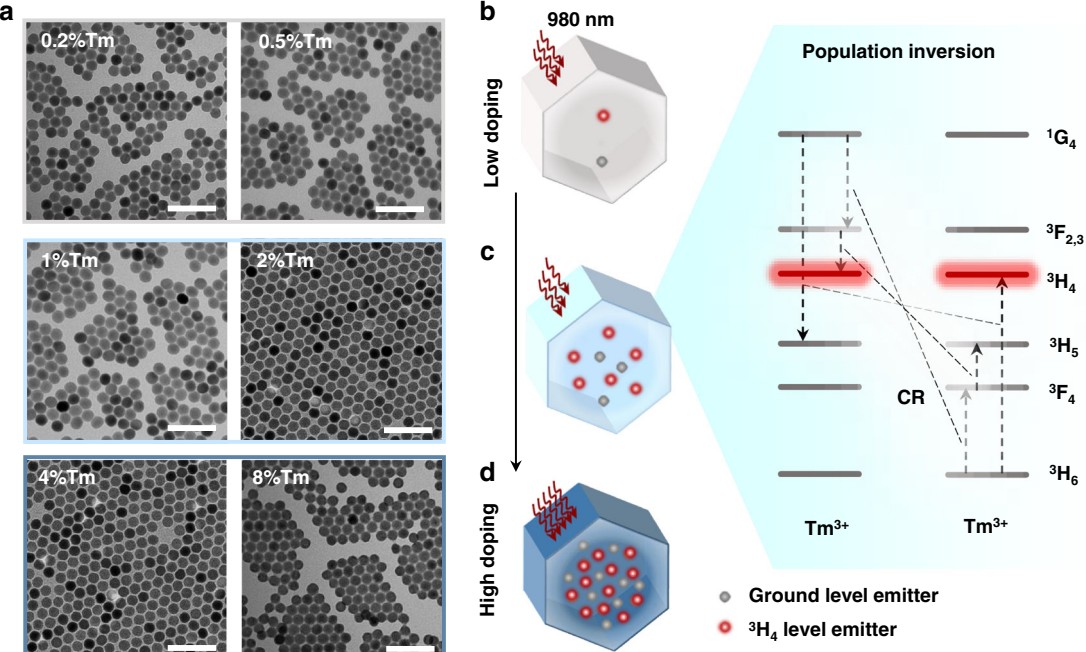

**Fig. 1 The role of CR in promoting the establishment of population inversion for low threshold lasing. a** TEM images of a series of monodispersed 24 nm UCNPs. Scale bar 100 nm. **b–d** Schematic illustration of the simplified energy level diagram and the role of doping concentration of $Tm^{3+}$ in establishing the CR enabled population inversion of $^3H_4$ level, compared to the ground level.

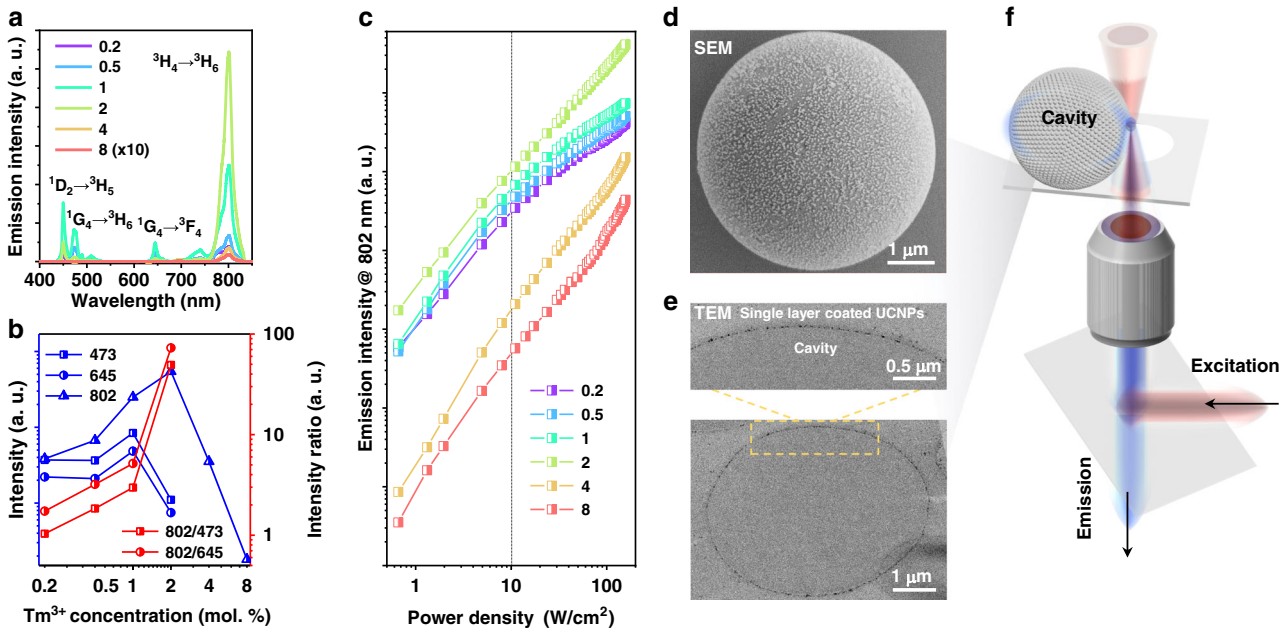

**Fig. 2 Power dependent emission characteristics of upconversion nanoparticles with tunable doping concentrations and self-assembled UCNPs as the gain medium on a polystyrene microsphere as the micro cavity. a** Comparison emission intensities and spectra of UCNPs doped with 20 mol.% $Yb^{3+}$ and x mol.% $Tm^{3+}$ (x = 0.2, 0.5, 1, 2, 4, 8). **b** Comparison emission intensities at the characteristic wavelengths and the intensity ratios of 802 nm/473 nm and 802 nm/645 nm. Both results displayed in (**a**, **b**) were acquired at the excitation power density of 100 W/cm². **c** Power dependent emission intensity evaluation at 802 nm. **d** SEM image of a 5-μm polystyrene micro cavity coated with a single layer of self-assembled UCNPs. **e** Cross-sectional TEM image of micro cavity showing the distribution of UCNPs on the surface. **f** Excitation and detection scheme (see Supplementary Fig. 5 for the details of a purpose-built imaging system).

and cross-sectional TEM measurements of a 5-μm polystyrene microsphere, showing the single layer of 2 mol.% $Tm^{3+}$ doped UCNPs on the surface of the cavity through the electrostatic force assembly (see zeta potential data in Supplementary Fig. 9). Microsphere confines the light through total internal reflection

and supports both transverse electric (TE) and transverse magnetic (TM) propagating modes with high quality (Q) factor, e.g., the Q factor for a 750 μm fused-silica microsphere could even reach ~$10^9$ [29–32]. When nanoparticles are coated onto the surface, the increased roughness induces scattering losses and reduces Q

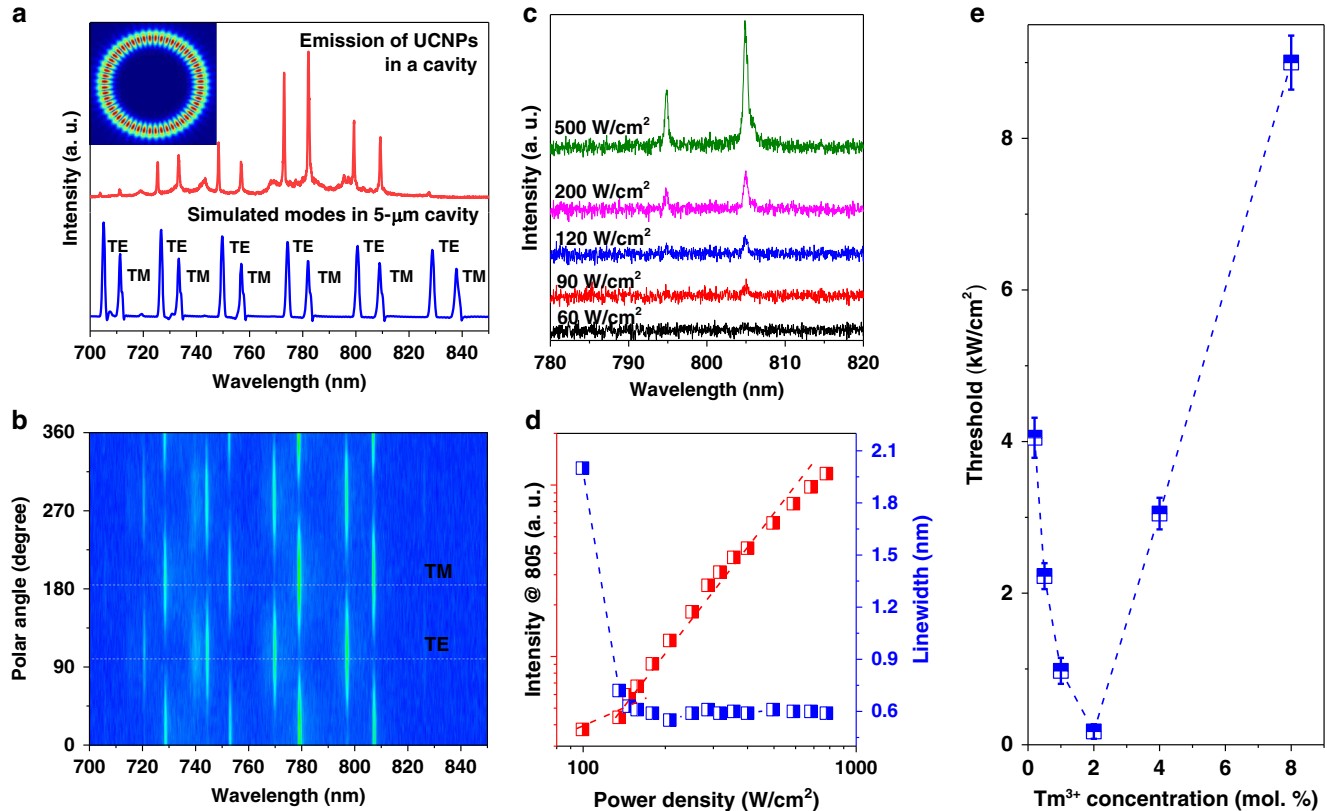

**Fig. 3 Characteristics of low threshold upconversion lasing emissions from a single layer of self-assembled UCNPs in a polystyrene microsphere. a** Numerical simulations of resonance spectrum and experimental emission spectrum. The inset shows a numerical simulation of the electrical-field distributions at 800 nm within a major plane. **b** Emission polarization angle-dependent intensities of the lasing peaks. **c** Power dependent upconversion emission spectra showing the gradual appearance of lasing peaks. **d** The pumping power-dependent plots of emission intensities and spectral linewidth narrowing, showing the onset of upconversion lasing emissions. **e** $Tm^{3+}$concentration-dependent thresholds for the onsets of upconversion lasing emissions.

factors[33], therefore homogeneous coatings avoid aggregations and minimize scattering.

Figure 3a shows the multi-mode lasing emissions under the pumping power of 10 kW/cm² and simulated optical modes of the cavity at 800 nm (see the details of Finite difference time domain simulations in Supplementary information and Supplementary Figs. 10 and 11). The 5-μm microsphere with a refractive index of ~1.59 leads to an effective coupling of upconversion emissions within the cavity modes. The full width at half-maximum (FWHM) of the sharp lasing peaks is calculated to be 0.54 nm using the Lorentz function fitting. According to $Q \approx \lambda_0/\Delta\lambda$, where $\lambda_0$ and $\Delta\lambda$ are the centre wavelength and FWHM of the peak profile, the quality factor Q is estimated as ~1500. Both the narrow FWHM and high Q benefit from the small size of ca. 24 nm of the UCNPs and the homogenous coating of a single layer of UCNPs (as the contrast, see the results of uneven coating in Supplementary Fig. 12). The inset in Fig. 3a and Supplementary Fig. 10 indicate that the jacinth-shaded region with high field intensity overlaps the coating layer of UCNPs.

The characterized polarization properties of both the TE and TM mode emissions exhibiting linear polarization with orthogonal periodicity as observed by the different polarization angles collection in Fig. 3b. The free spectral range (FSR) between two adjacent TE modes or TM modes fit well with the theoretical value ($\Delta\lambda_{FSR} = \frac{\lambda^2}{2\pi R}$, R is the radius of microcavity). Noting that the slight difference of experimental lasing peaks is attributed to the imperceptible slight variations of each cavity, as the mode position will move more than 8 nm when the size of this cavity changes for only 1% (Supplementary Fig. 11).

The transition from below, near, and at the threshold lasing emissions are seen in Fig. 3c when increasing the pumping powers to the onset of lasing emissions with the characteristic sharp and regularly spaced emission peaks. The narrow peaks that indicative of increased coherence and laser emissions are emerging above the threshold at pump intensities of 120 W/cm² for the laser mode at 805 nm. Both the slopes for the intensity and linewidth curves (Fig. 3d) display a non-linear change characteristic at the ultra-low pumping threshold of ~150 W/cm² achieved for the homogeneously coated 2 mol.% $Tm^{3+}$ doped UCNPs sample. Figure 3e shows 2 mol.% $Tm^{3+}$ doped UCNPs as the most effective gain medium as a result of the CR induced efficient population inversion at low pumping power (for the comparison characterizations of other UCNPs samples see Supplementary Fig. 13). Note that the accurate laser threshold should be ideally determined by using the absorbed pump power instead of incident pump power. We reason that the actual laser thresholds should be much lower than the current values once the absorbed power could be measured, though it's inaccessible in our experimental conditions.

Although the larger nanoparticles often produce stronger brightness, the size-induced scattering losses cannot be neglected. As shown in Fig. 4a–d, with the increase of nanoparticle size, from 24 nm, to 43 nm and 51 nm the FWHM in the lasing spectra increases from ~0.5 nm to 0.8 nm and ~1.2 nm, with the increased spontaneous emission background (Fig. 4e). Though larger sized UCNPs are brighter due to the reduced degree of surface-quenching, by testing more than 20 micro cavity lasers, the Q factor reduces from ~1900 to ~700 with the increased size of UCNPs (Fig. 4f). The reduction in the Q factor indicates the

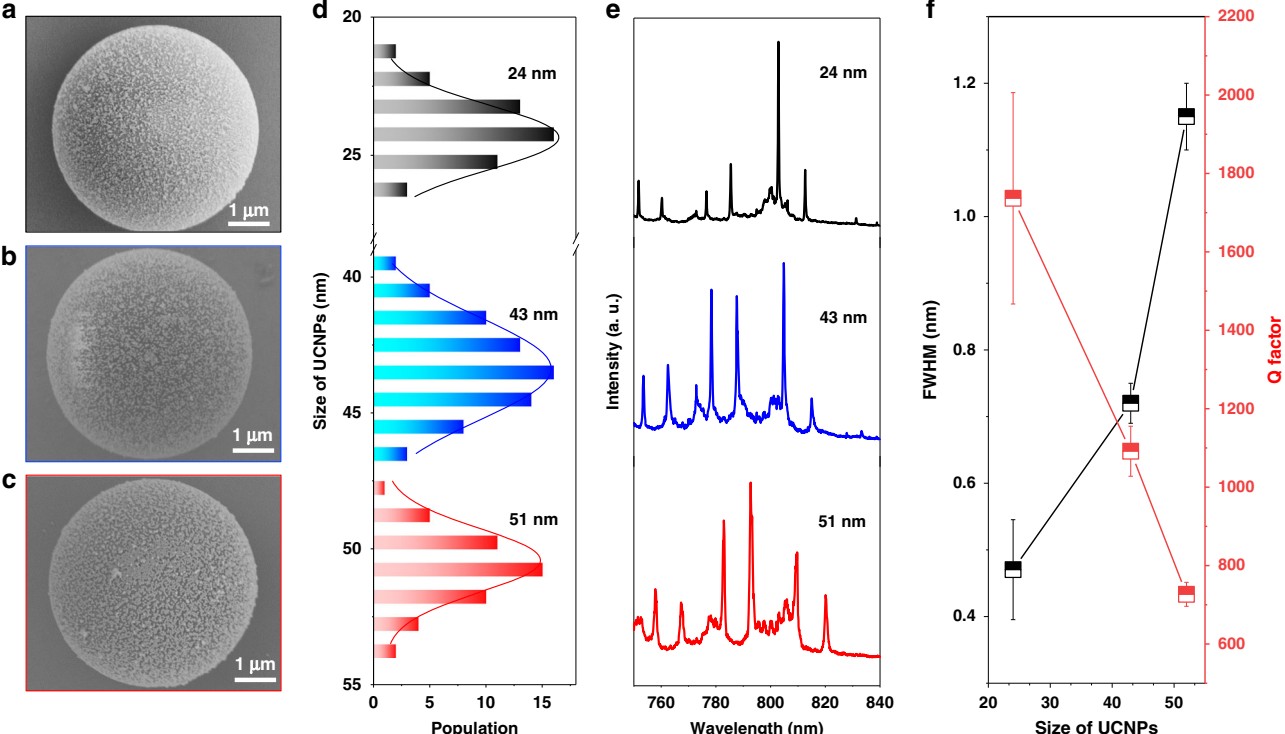

**Fig. 4 Size-dependent quality (Q) factors caused by the scattering losses when coating a single layer of nanoparticles in the micro cavity. a–c** SEM images of microspheres coated with $NaYF_4$:20%$Yb^{3+}$,2%$Tm^{3+}$ nanoparticles with sizes of 24 nm, 43 nm, and 51 nm. **d** Size distributions of UCNPs used in (**a–c**). Related **e** emission spectra **f** FWHM and Q factor of UCNPs coated micro cavities in (**a–c**).

increased size-induced scattering loss. When the size of UCNPs becomes too large, the catchment area by the evanescent field of micro cavity modes also reduces (Supplementary Figs. 10 and 14).

**Characterization of single nanocrystal lasing**. Figure 5a, b further show that lanthanide ions doped single UCNPs are optically uniform and sufficiently bright, as there are averaged 23640 ± 832 photon counts per 50 milliseconds from single nanoparticles under a confocal microscoy, which are suitable for single particle lasing in Fig. 5c. We demonstrate that CR assisted population inversion in single UCNPs as a highly efficient gain medium, which can be widely used in any photonics cavities for single nanoparticle lasers. Single 43 nm $NaYF_4$: 20 mol.% $Yb^{3+}$-2 mol.% $Tm^{3+}$ UCNPs can lase at 808 nm with the sharp peaks and narrow FWHM of ~0.45 nm once in a cavity (Fig. 5c–f), which preserves the Q-factors of the cavity since scattering losses are reduced to the minimum. Due to the reduced scattering from single UCNP, it allows to use of bigger particles and increase the total gain, compared to the uniform 24 nm single layer coating.

**Conclusion**. We have achieved ultra-low threshold lasing by controlling CR in an upconversion energy transfer system and thereby the easy establishment of population inversion in single UCNPs. By using the doping-concentration-optimized UCNPs as the gain medium and single layer coating of monodispersed UCNPs on the cavity with minimized scattering loss, strong absorption and high efficiencies for both upconversion and population inversion establishment have been achieved. Employing a 5-μm cavity and homogenous coating architecture, we have achieved lasing emissions with a threshold of ~150 W/cm², nearly two orders of magnitude lower than the recently reported benchmark value of 14 kW/cm². We have further verified that the size of single nanoparticle does not affect the Q factor of micro cavity lasing, with an FWHM of ~0.45 nm achieved for

upconversion lasing emissions from a single 43 nm nanoparticle. This study suggests great potential to using the concentration tunable UCNPs as an efficient gain medium for room temperature CW microscale and nanoscale lasers. The single upconversion nanocrystal lasing offers prospects to achieve NIR pumped anti-Stokes nanolaser platform for a variety of practical applications, such as intracellular tagging and imaging[34]. A further reduction of the threshold values and mode volumes is achievable by optimizing the combination strategy between the gain medium and cavity, and selecting higher-Q cavities, e.g., plasmonic nanocavity, Spaser, hyperbolic metacavity, photonic crystals or photonic topological insulator array cavities[35,36]. The CR mediated lanthanides energy transfer system might be embedded in semiconducting matrixes to achieve electrical pumped emission and further expand low threshold CW lasers via electrical pumping, which are more compatible with current standard technologies[37].

## Methods

**Upconversion nanoparticles synthesis**. The uniform oleic-acid-capped nanoparticles (β-$NaYF_4$:20%Yb, $x$%$Tm^{3+}$) were synthesized by the coprecipitation method. A typical procedure is as follows: $YCl_3·6H_2O$ (0.8-$x$ mmol), $YbCl_3·6H_2O$ (0.2 mmol) and $TmCl_3·6H_2O$ ($x$ mmol) were added into a 50 mL three-necked flask containing 6 mL oleic acid (OA) and 15 mL 1-octadecene (ODE). The mixture was first heated to 160 °C under argon for 30 min to form a transparent solution and remove residual water. The solution was cooled down to room temperature, and 10 mL of a methanol solution containing NaOH (2.5 mmol) and $NH_4F$ (4 mmol) was slowly dropped into the flask and stirred for 30 min. Then, the solution was heated to 70–80 °C and maintained for 30 min to evaporate methanol. Subsequently, the solution was heated to 300 °C and maintained for 1 h under argon atmosphere. After cooling down to room temperature, the resulting products were precipitated by ethanol and collected by centrifugation at 6000 rpm for 5 min. The precipitate was then purified with ethanol three times, and finally dispersed in cyclohexane for further use.

**Surface modification for water-soluble upconversion nanoparticles**. In a typical procedure, the nanoparticles were first precipitated by adding ethanol (2.0 mL) to a cyclohexane colloidal solution of the OA coated hydrophobic β-

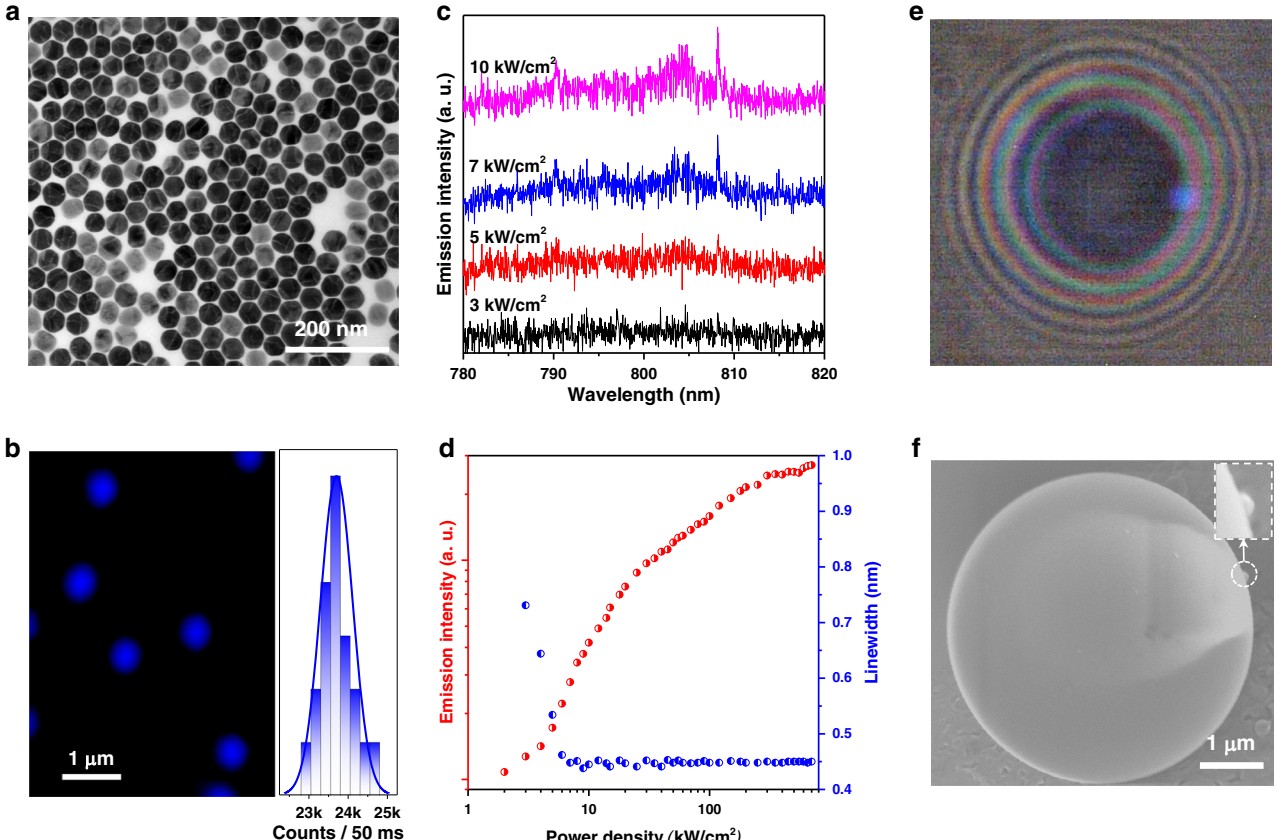

**Fig. 5 Lasing emissions from a single nanocrystal. a** TEM image of 43 nm monodispersed NaYF$_4$:20%Yb$^{3+}$,2%Tm$^{3+}$ nanocrystals. **b** Point-scanning confocal microscopic image of single UCNPs (in pseudocolour) and their brightness distribution at an excitation power density of 100 kW/cm$^2$. **c** Power dependent upconversion emission spectra showing the gradual appearance of lasing emission peaks at 808 nm. **d** The pumping power-dependent plots of emission intensities and spectral linewidth narrowing (peak@808 nm), showing the onset of single UCNP's lasing emissions. **e**, **f** Wide field image (**d**) and SEM image (**e**) of a single UCNP in a micro cavity.

NaYF$_4$:20%Yb, $x$%Tm$^{3+}$ nanoparticles (1 mL, 10 mg/mL) and then collected by centrifugation at 6000 rpm for 5 min. The obtained nanoparticles were re-dispersed in a mixed solution of ethanol (1 mL) and HCl (1 mL, 2 M) upon sonication for 5 min. The ligand free nanoparticles were collected by centrifugation at 14680 rpm for 10 min and re-dispersed in deionized water (1 mL).

**Fabrication of upconverting microlasers.** Upconverting microlasers were produced through a solution process. In a typical procedure, 10 μL PS microbeads (10% solids) were first dispersed in 1000 μL DI water upon sonication for 5 min, then 7 μL ligand free water-soluble UCNPs (8 mg/mL) were mixed together with PS microbeads upon sonication. Then, this mixed solution was kept for a gentle shake (750 rpm) for 2 h before centrifugation. These UCNPs coated microbeads were further washed with ethanol and water for three times, and finally were re-dispersed in water. Noting that, the coated surface and thickness can be controlled precisely by tuning the concentration ratio of microbeads and nanoparticles.

**Characterization and simulation**

*TEM*. Transmission electron microscope (TEM) measurements were performed using a FEI Tecnai T20 instrument with an operating voltage of 120 kV. The samples for TEM analysis were prepared by placing a drop of a dilute suspension of nanoparticles onto carbon-coated copper grids. For the cross-sectional TEM images, we first embedded the single layer UCNPs coated microcavity in epoxy (353ND), and then cut the microspheres using Leica microtome to get ultrathin slices.

*SEM*. The morphology of microcavity coated with UCNPs was characterized via scanning electron microscope (SEM) imaging (Supra 55VP, Zeiss) operated at 3.00 kV.

*Zeta potential*. The measurement of zeta potential (ζ-potential) was carried out by Zetasizer (Malvern Panalytical).

**Confocal microscopy and spectroscopy.** The inverted confocal optical system was built on a sample scanning configuration employing a 3D piezo stage. A single-mode fiber-coupled 980 nm diode laser was used as the excitation source and was

focused onto the sample through an oil-immersion objective lens (UPlanAPO, Olympus; ×100, NA = 1.4). The emission from sample was collected by the same objective lens then refocused into an optical fiber that has a core size matching with system Airy disk. A single photon counting avalanche diode (SPAD) detector was connected to the collection optical fiber to detect the emission intensity. The spectra were measured with a fiber-coupled spectrometer (Andor) with a grating of 1200 grooves/mm (resolution: 0.21 nm, data collection step: ~0.04 nm). And the polarization performance of this upconverting microlaser was characterized by rotating a λ/2 plate @808 nm while cooperated with a polarizer (620–1000 nm).

**Finite difference time domain simulations**. Numerical simulations of our microcavity were performed using Lumerical FDTD solutions. For the simulation of the electrical-field distributions and resonance spectrum, the perfectly matched layer (PML) was set as the boundary condition with a simulation region of 6 μm × 6 μm × 6 μm; the monitored wavelength was 400–1000 nm; the meshes order was 10 nm; the light type was a dipole source. The simulated resonance spectrum and electric field distribution of microcavity were calculated separately, as shown in Fig. 3a. The cavity modes in Supplementary Fig. 11 were achieved while changing the sizes of the microcavity. The electrical-field distributions at ~800 nm mode was plotted along the diameter.

## Data availability

All the relevant data are available from the corresponding authors upon reasonable request.

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

## Acknowledgements

This project is primarily supported by ARC Discovery Early Career Researcher Award Scheme (J.Z., DE180100669), China Scholarship Council Scholarships (Y.S.: No. 201706120322), National Natural Science Foundation of China (NSFC, 61729501), Major International (Regional) Joint Research Project of NSFC (51720105015), Science and Technology Innovation Commission of Shenzhen (KQTD20170810110913065), and Australia China Science and Research Fund Joint Research Centre for Point-of-Care Testing (ACSRF658277, SQ2017YFGH001190).

## Author contributions

Y.S., J.Z., and D.J. conceived the project and designed experiments. J.Z., C.Y., L.J., and D.J. supervised the project. Y.S. and Y.C. synthesized the upconversion nanoparticles. Y.S., F.W., and J.Z. conducted the optical setup. Y.S. fabricated the microlasers and performed the characterizations. Y.S., J.Z., A.F., and D.J. analyzed the results, prepared the figures and wrote the paper. All authors contributed to discussion, interpreting the data and the writing.

## Competing interests

The authors declare no competing interests.
