## [Peer Review File · Nature Communications]

REVIEWER COMMENTS

Reviewer #1 (Remarks to the Author):

In this manuscript, lanthanides doped upconversion nanoparticles (UCNPs) were coated on the surface of a 5- μm -diameter polystyrene microsphere to implement an up-converted laser. The cross-relaxation (CR) was optimized through doping of Tm^{3+} ions.

However, this work is quite similar to ref. 23. Furthermore, the authors didn't show lasing characteristics clearly, particularly in the microcavity with a single nanocrystal.

Detailed comments:

- (1) Similar microlasers using UCNPs have been already reported in ref. 23. It is not clear what is the difference between these two studies. Also, it was not well explained in this work why the threshold was an order of magnitude lower than that in ref. 23.
- (2) Figure 2d shows a SEM image of a 5- μm polystyrene microcavity coated with UCNPs. But, the authors should show a clear evidence of single layer coating of UCNPs (e.g. TEM image).
- (3) More importantly, the authors should show clear lasing behaviors. For example, the above-threshold spectrum was not clearly shown in Fig. 3c. Since the threshold was $\sim 150 \text{ W/cm}^2$ in light in-light out curve (Fig. 3d), it is necessary to show a spectrum measured at the pump power much larger than 200 W/cm^2 . In addition, lasing thresholds were not clearly shown in light in-light out curves in Fig. S9 (as well as Fig. 3d). I suggest that the authors show these curves in log-log plot.
- (4) In Fig. 4, the authors estimated Q factors using the FWHM in the lasing spectra to understand the scattering loss depending on the nanoparticle size. However, an estimate of the Q factor should be made by measuring the linewidth of the resonance at transparency (near or below threshold), not by using the laser linewidth [IEEE J. Sel. Topics in Quant. Electron. 5, 673 (1999)]. The laser linewidth is generally determined by many factors including Q factor and optical gain, not by only Q factor.
- (5) The microcavity with a single nanocrystal does not show lasing characteristics at all (Fig. 5). In particular, the authors didn't show light in-light out curve and above-threshold spectrum to convince lasing. Small peaks at $\sim 800 \text{ nm}$ seem to be typical resonances below threshold (Fig. 5c).
- (6) A 5- μm -diameter microsphere with a refractive index of ~ 1.59 cannot achieve a Q factor of 109, although the authors seemed to expect such a high Q factor (page 5). In addition, the authors mentioned that the field overlap between UCNPs and whispering-gallery mode would be high (Fig. 3a). But, I don't think so because UCNPs was coated on the outer surface of the microsphere. It would be necessary to calculate quantitatively the field overlap.
- (7) Was the CCD image in Fig. 2e shown using a 800-nm wavelength filter? Why was the color of the image blue? Also, why was only red color emitted in the image of Fig. 5b, although the emission at 450 nm was strong in Fig. 5c? Furthermore, why was the spectrum of Fig. 5c different from Fig. 2a and b (i.e. why was the ratio of 450 nm vs. 800 nm light different)?

Reviewer #2 (Remarks to the Author):

In this work authors demonstrate how the threshold of a microsphere laser composed of a thin layer of UCNPs growth on a microsphere resonator can be improved by adequate doping. The developed structures are of super high quality and their performance as optical amplifiers has been systematically investigated. The results are timely as this is an emerging field. The impact of these results in the field could be very high.

Overall I think this is a very nice piece of work. Nevertheless, I have some major concerns. These are explained next. Note that the main one is the correlation of the existence of an optimum Tm content with the tuning of the cross relaxation.

- My main concern when reading the paper is about the fact that the threshold reduction observed by authors, a very valuable result, is attributed solely to the concentration induced modulation of the cross relaxation. Authors are determining the laser threshold from the intensity and linewidth curves versus the incident 980 nm laser. Accurate determination of laser threshold requires the plot of the same curves as a function of the absorbed pump power instead of incident pump power. This is not possible in their experimental conditions but I think then the conclusion should be discussed further. The laser threshold reduction can be also caused by an increase in the efficiency of the Tm excitation efficiency caused by an increment in the Yb-Tm energy transfer. It is known that the Yb to Tm energy transfer efficiency increases with concentration due to the reduction in the Yb-Tm distance. At very large Tm concentrations, quenching and Tm-Yb energy back transfer could take place. It results that at low Tm concentrations the Yb-Tm energy transfer increases with Tm concentration and at very large Tm concentrations the Yb-Tm effective energy transfer decreases with the Tm concentration. This effect would also lead to the appearance of an optimum Tm concentration for the laser threshold. This should be discussed in detail. Authors should estimate the overall Yb-Tm energy transfer efficiency as a function of Tm concentration. It would be also interesting to have the dependence of Yb absorption and lifetime on the Tm concentration.
- Authors state: "As the CR dynamics also depend on the excitation power.." Why is that? This should be explained in more detail.
- Why Figure 2e evidences the presence of WGMs? The appearance of an opposite bright spot would occur only for certain WGMs with wavelength equal to the diameter of microsphere. Is this the case? which is the order of the WGMs in this case (it can be estimated from emission spectra).
- The discussion related to Figure 2b would benefit from the inclusion of the ratio between bands and how it depends on the Tm concentration.
- In the laser curves included in the SI the presence of a laser threshold is not so evident as in the data authors included in the main text. Why is this? How authors can be sure that what they are observing is laser emission instead of a spectral modulation of the spontaneous emission due to the presence of WGMs? Can authors measure the lifetime as a function of pump power to check for the stimulated emission induced lifetime shortening?

Reviewer #3 (Remarks to the Author):

In their manuscript Zhou et al. report on lasing emissions from single NaYF₄:Yb,Tm nanocrystals as part of a microcavity. By optimizing the Tm³⁺ concentration, they achieved upconverted laser emission at room temperature upon cw excitation with only 150 W/cm². This low lasing threshold is quite remarkable as such but also because it was achieved without reducing surface quenching of the emission by coating the nanocrystals with an inert shell. Their work significantly expands the potential of using upconverting nanocrystals for microscale lasers. The nanomaterials have been well characterized, all conclusions are supported by the data and the figures are of appropriate quality. I therefore support publication of the manuscript in Nature Communications without any modifications.

Response Letter

Our point-by-point responses (in blue) to the reviewers' comments (in black) are provided below.

Reviewer #1 (Remarks to the Author):

In this manuscript, lanthanides doped upconversion nanoparticles (UCNPs) were coated on the surface of a 5- μm -diameter polystyrene microsphere to implement an up-converted laser. The cross-relaxation (CR) was optimized through doping of Tm^{3+} ions.

However, this work is quite similar to ref. 23. Furthermore, the authors didn't show lasing characteristics clearly, particularly in the microcavity with a single nanocrystal.

Response: The focus of this work is to investigate the upconversion nanoparticles as the efficient gain medium on a microcavity, which allows us to achieve continuous-wave upconversion laser with the threshold as low as $\sim 150 \text{ W/cm}^2$. The distinctions between this work and ref. 23 have been analyzed in our response to this reviewer's comment (1).

To clearly show the lasing characteristics, particularly in the microcavity with a single nanocrystal, we have performed more measurements.

Detailed comments:

(1) Similar microlasers using UCNPs have been already reported in ref. 23. It is not clear what is the difference between these two studies. Also, it was not well explained in this work why the threshold was an order of magnitude lower than that in ref. 23.

Response: This work studies the commonly used $\text{Yb}^{3+}/\text{Tm}^{3+}$ codoped UCNPs for low-threshold lasing, and the results reported in ref 23 were achieved by the Tm^{3+} singly doped UCNPs. The difference in the doping lead to the differences in population inversion mechanism, upconversion energy transfers, and absorption capability. Specifically:

- 1) As shown in Fig. R1a, the cross-relaxation in the $\text{Yb}^{3+}/\text{Tm}^{3+}$ codoped UCNPs facilitates the population inversion of $^3\text{H}_4$ level of Tm^{3+} upon the 980 nm excitation, while the avalanche-like energy looping in Tm^{3+} singly doped UCNPs plays the essential role in establishing the population inversion upon the 1064 nm excitation in ref. 23 (Fig. R1b).

Figure R1. Energy level diagrams showing the mechanism for Tm^{3+} population inversion. **a**, $\text{Yb}^{3+}/\text{Tm}^{3+}$

co-doped UCNPs having typical cross-relaxation among Tm^{3+} emitters under the excitation of 980 nm, which accumulates the population at Tm^{3+} : $^3\text{H}_4$ excite state. **b**, Tm^{3+} singly doped nanoparticles under the excitation of 1064 nm involving an energy loop process to facilitate the population inversion of Tm^{3+} : $^3\text{H}_4$ excite state.

- 2) The upconversion mechanism, efficiency and brightness of the two gain media are also quite different. The upconversion luminescence of $\text{Yb}^{3+}/\text{Tm}^{3+}$ co-doped UCNPs under 980 nm excitation relies on the energy transfer upconversion (ETU) mechanism, while Tm^{3+} doped UCNPs under 1064 nm excitation requires excited state absorption (ESA).
- 3) The absorption cross-section of Yb^{3+} at 980 nm ($\sim 10^{-20} \text{ cm}^2$, ground-state absorption, $^2\text{F}_{7/2} \rightarrow ^2\text{F}_{5/2}$)^{1,2} is more than one order of magnitude larger than that of Tm^{3+} at 1064 nm ($\sim 10^{-21} \text{ cm}^2$, excited-state absorption, $^3\text{F}_4 \rightarrow ^2\text{F}_{2,3}$)³.
- 4) Besides, in our study, we achieved a single layer of the gain medium on the surface of a microcavity, which maintains the cavity with a higher Q factor and negligible scattering loss.

All of the above differences led to the one order of magnitude lower lasing threshold achieved in our work, i.e. 1-3) contribute to the population inversion condition at low pumping power and 4) promotes the signal amplification process.

To highlight these synergistic contributions to the much lower threshold, we have added in the revised manuscript, in lines 200-203, Page 9:

“By using the doping-concentration-optimized UCNPs as gain medium and single-layer coating of monodispersed UCNPs on the cavity with minimized scattering loss, strong absorption and high efficiencies for both upconversion and population inversion establishment have been achieved.”

(2) Figure 2d shows a SEM image of a 5- μm polystyrene microcavity coated with UCNPs. But, the authors should show a clear evidence of single layer coating of UCNPs (e.g. TEM image).

Response: Thanks for this very constructive suggestion. To provide clear evidence of homogeneous single-layer coating, we both zoomed in the SEM image and performed a TEM cross-section measurement, and added the new results in the supplementary information as Figure S8.

Figure S8 **a**, Enlarged SEM image of microcavity surface; **b**, TEM image of epoxy embedded microtome cross-section (slice thickness: $\sim 50 \text{ nm}$) of microcavity coated with a single layer of UCNPs. All the UCNPs are coated on the outer surface of the microsphere.

(3) More importantly, the authors should show clear lasing behaviors. For example, the

above-threshold spectrum was not clearly shown in Fig. 3c. Since the threshold was $\sim 150 \text{ W/cm}^2$ in light in-light out curve (Fig. 3d), it is necessary to show a spectrum measured at the pump power much larger than 200 W/cm^2 . In addition, lasing thresholds were not clearly shown in light in-light out curves in Fig. S9 (as well as Fig. 3d). I suggest that the authors show these curves in log-log plot.

Response: Thanks for this constructive suggestion. Following this reviewer's advice, we have added a spectrum data measured at 500 W/cm^2 in Figure 3c. We have accordingly revised the power-dependent curves in Figure 3d and Figure S13 in a log-log plot.

Figure 3c. Power dependent upconversion emission spectra showing the gradual appearance of lasing peaks. **d.** The pumping power-dependent plots of emission intensities and spectral linewidth narrowing (peak@805 nm), showing the onset of upconversion lasing emissions.

Figure S13 The systematic characterizations of the lasing thresholds for the cavities coated with UCNPs at Tm^{3+} doping concentrations of 0.2%, 0.5%, 1%, 4%, 8%.

(4) In Fig. 4, the authors estimated Q factors using the FWHM in the lasing spectra to understand the scattering loss depending on the nanoparticle size. However, an estimate of the Q factor should be made by measuring the linewidth of the resonance at transparency (near or below threshold), not by using the laser linewidth [IEEE J. Sel. Topics in Quant. Electron. 5, 673 (1999)]. The laser linewidth is generally determined by many factors including Q factor and optical gain, not by only Q factor.

Response: Thanks this reviewer for bringing this issue for further discussions. Indeed, the total resonance linewidth is determined by multiple loss possibilities, including radiation, surface scattering, material absorption, and others. Accordingly, the total (or loaded) quality factor Q_{tot} ($\approx \lambda_0/\Delta\lambda$) can be written in the form of $Q_0^{-1} = Q_{\text{rad}}^{-1} + Q_{\text{sca}}^{-1} + Q_{\text{mat}}^{-1} + \dots$ ⁴. The Q_{sca} and Q_{mat} indicators result from the optical gain, while in our case the scattering loss dominates. It is a controversial issue to estimate the Q factor by using the linewidth near or below threshold, since Q factors of a passive and active resonator can be different and are also power dependent. Our the performance of the gain media in our case non-linearly depends on the excitation power, which ultimately affects the overall Q-factor. However, the measured linewidth around threshold doesn't show much difference as the power reaches the threshold. Therefore, it is reasonable to reveal the size dependent scattering loss using linewidth.

To avoid misunderstanding, we have revised the description in the main text as below:

Lines 144-145, Page 6 “According to $Q \approx \lambda_0/\Delta\lambda$, where λ_0 and $\Delta\lambda$ are the centre wavelength and FWHM of the peak profile, the quality factor Q is estimated as ~ 1500 .”

Lines 175-179, Page 8 “Though larger sized UCNPs are brighter due to the reduced degree of surface-quenching, by testing more than 20 micro cavity lasers, the Q factor reduces from ~ 1900 to ~ 700 with the increased size of UCNPs (Figure 4f). The reduction in the Q factor indicates the increased size-induced scattering loss. When the size of UCNPs become too large, the catchment area by the evanescent field of micro cavity modes also reduces (Figure S10 and S14).”

(5) The microcavity with a single nanocrystal does not show lasing characteristics at all (Fig. 5). In particular, the authors didn't show light in-light out curve and above-threshold spectrum to convince lasing. Small peaks at ~ 800 nm seem to be typical resonances below threshold (Fig. 5c).

Response: Follow this reviewer's suggestion, we have measured the power dependent spectra of the single nanocrystal's lasing emissions with the results being included in the revised Figure 5c (the previous Figure 5c became Figure S15). We have also plotted the light in-light out curves and power-dependent linewidth narrowing curve in Figure 5d.

Figure 5c, Power dependent upconversion emission spectra showing the gradual appearance of lasing emission peaks at 808 nm. **d**, The pumping power-dependent plots of emission intensities and spectral linewidth narrowing (peak@808 nm), showing the onset of single UCNP's lasing emissions.

(6) A 5- μm -diameter microsphere with a refractive index of ~ 1.59 cannot achieve a Q factor of 10^9 , although the authors seemed to expect such a high Q factor (page 5). In addition, the authors mentioned that the field overlap between UCNPs and whispering-gallery mode would be high (Fig. 3a). But, I don't think so because UCNPs was coated on the outer surface of the microsphere. It would be necessary to calculate quantitatively the field overlap.

Response: Thank this reviewer for very constructive comments here.

We apologize for our misleading message. The Q factor of 10^9 should have referred to WGM cavity based on 750 μm fused-silica microsphere at 633 nm in the refs. 29 and 32, not the 5 μm -diameter PS microsphere in this work. Accordingly, we have revised the sentence to:

Lines 121-1214, Page 5 “*Microsphere confines the light through total internal reflection and supports both transverse electric (TE) and transverse magnetic (TM) propagating modes with high quality (Q) factor; e.g. the Q factor for a 750 μm fused-silica microsphere could even reach $\sim 10^9$.*”

We have also followed this reviewer's suggestion and calculated the degree of field overlap. In our design, UCNPs were coated on the outer surface of the microsphere. Though the area with strongest field intensity appear at the region slightly inside the cavity, embedding nanoparticles inside the cavity will destroy the surface smoothness of the cavity and reduce the Q factor. To clarify this configuration, we have enlarged the field simulation figure in Figure S10 to show the overlap and also added the following description in the revised manuscript:

Lines 147-148, Page 6 “*The inset in Figure 3a and Figure S10 indicate that the jacinth-shaded region with high field intensity overlaps the coating layer of UCNPs*”.

Figure S10 Numerical simulation of the electrical-field distributions at 800 nm (eigenmode) within a major plane.

(7) Was the CCD image in Fig. 2e shown using a 800-nm wavelength filter? Why was the color of the image blue? Also, why was only red color emitted in the image of Fig. 5b, although the emission at 450 nm was strong in Fig. 5c? Furthermore, why was the spectrum of Fig. 5c different from Fig. 2a and b (i.e. why was the ratio of 450 nm vs. 800 nm light different)?

Response: The image in Figure 2e was measured using a sCMOS camera (Prime 95B) with a 800 nm bandpass filter. It is a pseudocolour image from the intensity image processed by imageJ. To avoid the confusion of blue emissions from 450 nm and 475 nm, we have revised this image to two-colour gradient diagram and marked this as pseudocolour in the figure caption (see new Figure 2e).

Figure 2e Wide-field pseudocolour image (from the 800 nm-bandpass intensity image processed by imageJ) of a lasing micro cavity pumped by the laser excitation of a diffraction-limited spot on the side of the microsphere.

Figure 5b is a point-scanning confocal image with the emission intensity collected by a single-photon counting avalanche diode from 400 nm to 850 nm. This type of confocal images have been routinely used to characterize the absolute brightness and intensity uniformity of single nanoparticles (see refs

2, 14, 15, 16). All the confocal image results were presented in pseudocolour (now revised in blue in the current Figure 5b).

Figure 5b Point-scanning confocal microscopic image of single UCNPs (in pseudocolour) at an excitation power density of 100 kW/cm².

The difference of dramatic intensity ratios in the spectra of previous Figure 5c and Figure 2a was due to the different excitation power densities used (100 kW/cm² v.s. 100 W/cm²). The 800 nm emission comes from a 2-photon populated upconversion process while the 450 nm emission follows a 4-photon process. They have the power dependence with square and biquadrate relationships ($I \propto p^2$ and $I \propto p^4$), respectively. At a low power density, 800 nm emission dominates the spectrum, while at a high power density, ³H₄ level (800 nm) becomes saturated and the energy will be up populated to the higher excited states (e.g., ¹G₄ level, produces 450 nm emission) through the multi-photon process.

Reviewer #2 (Remarks to the Author):

In this work authors demonstrate how the threshold of a microsphere laser composed of a thin layer of UCNPs growth on a microsphere resonator can be improved by adequate doping. The developed structures are of super high quality and their performance as optical amplifiers has been systematically investigated. The results are timely as this is an emerging field. The impact of these results in the field could be very high.

Response: We thank this reviewer for his/her overall positive comments.

Overall I think this is a very nice piece of work. Nevertheless, I have some major concerns. These are explained next. Note that the main one is the correlation of the existence of an optimum Tm content with the tuning of the cross relaxation.

(1) Authors are determining the laser threshold from the intensity and linewidth curves versus the incident 980 nm laser. Accurate determination of laser threshold requires the plot of the same curves as a function of the absorbed pump power instead of incident pump power. This is not possible in their experimental conditions but I think then the conclusion should be discussed further. The laser threshold reduction can be also caused by an increase in the efficiency of the Tm excitation efficiency caused by an increment in the Yb-Tm energy transfer. It is known that the Yb to Tm energy transfer efficiency increases with concentration due to the reduction in the Yb-Tm distance. At very large Tm concentrations, quenching and Tm-Yb energy back transfer could take place. It results that at low Tm concentrations the Yb-Tm energy transfer increases with Tm concentration and at very large Tm concentrations the Yb-Tm effective energy transfer decreases with the Tm concentration. This effect would also lead to the appearance of an optimum Tm concentration for the laser threshold. This should be discussed in detail. Authors should estimate the overall Yb-Tm energy transfer efficiency as a function of Tm concentration. It would be also interesting to have the dependence of Yb absorption and lifetime on the Tm concentration.

Response: Thanks this reviewer for his/her great suggestions on the additional discussions on the doping concentrations.

It's true that accurate determination of laser threshold requires the value of absorbed pump power, which is inaccessible in our experimental condition. To discuss the effect from the incident pump power in determining the threshold, we have added the following description in Lines 163-166 Page 7:

“Note that the accurate laser threshold should be ideally determined by using the absorbed pump power instead of incident pump power. We reason that the actual laser thresholds should be much lower than the current values once the absorbed power could be measured, though it's inaccessible in our experimental conditions.”

As suggested, we have measured both the Yb³⁺ absorption and lifetime on the Tm³⁺ concentration. As shown in Figure S3, the Yb³⁺ absorbance keeps constant at different Tm³⁺ concentrations. However, as shown in Figure S4a, the Yb³⁺ lifetime decreases with the increase of Tm³⁺ concentrations, which indicate efficient energy transfer from Yb³⁺ to Tm³⁺ due to the shorted ions distance. We therefore estimated the overall Yb³⁺-Tm³⁺ energy transfer efficiency as a function of Tm³⁺ concentration, plotted in Figure S4b. The efficiencies increase with the increase of Tm³⁺ concentrations with a trend of saturation at the range of higher concentrations, which indicates the possible back energy transfer from Tm³⁺ to Yb³⁺, as predicted by this reviewer. This also contributes to the appearance of an

optimum Tm^{3+} concentration for the laser threshold.

We have added new Figure S3 and S4 and related analysis in the supplementary information, and the following discussion in lines 77-84, Page 3 in the main text:

“Besides, highly Tm^{3+} doped UCNPs leads to the decrease in the Yb^{3+} - Tm^{3+} distance and increase in the Yb^{3+} - Tm^{3+} energy transfer efficiency. But this strategy, as illustrated in Figure 1d, could lead to a quenching of the overall upconversion emissions, when doping too many emitters. As shown in Figure S4, the decrease in the Yb^{3+} lifetime values and the trend of energy transfer efficiency saturation appeared at the very large Tm^{3+} concentrations (e.g. > 8 mol.%), indicate the possible Tm^{3+} - Yb^{3+} energy back transfer²⁷. To alleviate the concentration quenching and energy back transfer effects in the highly doped UCNPs, high excitation power density is required to pump the significant amount of the ground level Tm^{3+} emitters and Yb^{3+} sensitizers.”

Figure S3 Absorption spectra of $\text{NaYF}_4:20\text{Yb}^{3+}/x\text{Tm}^{3+}$ ($x=0.2, 2, 8$).

Figure S4 a, Lifetime decay curves of Yb^{3+} emissions at 980 nm and **b**, Yb^{3+} to Tm^{3+} energy transfer efficiencies for UCNPs with different Tm^{3+} doping concentrations.

Note that we measured the absorption spectra (Figure S3) and lifetimes (Figure S4a) of Yb^{3+} on the increased concentrations of Tm^{3+} , and estimated the overall Yb^{3+} - Tm^{3+} energy transfer efficiencies (Figure S4b). Though the Yb^{3+} absorbance hasn't been affected, the Yb^{3+} - Tm^{3+} energy transfer efficiencies increase with Tm^{3+} concentrations with a trend of saturation, which indicates the possible minor effect induced by back energy transfer from Tm^{3+} to Yb^{3+} .

(2)- Authors state: "As the CR dynamics also depend on the excitation power." Why is that? This should be explained in more detail.

Response: This is because the population distribution of each energy level is strongly dependent on the excitation power density, due to the non-linear characteristics of the multi excited states of lanthanide ions [ref. 2, 3, 10, 14, 15, 16, 18]. We have added more details in the revised manuscript, see lines 110-112, Pages 4-5:

"As CR involves different energy levels between a pair of nearby emitters, the population distribution of each energy level is strongly dependent on the excitation power density, so is the CR dynamics."

(3)- Why Figure 2e evidences the presence of WGMs? The appearance of an opposite bright spot would occur only for certain WGMs with a wavelength equal to the diameter of microsphere. Is this the case? which is the order of the WGMs in this case (it can be estimated from emission spectra).

Response: As we excite the gain medium on the edge of this WGM cavity with a diffraction-limited spot, the light, which can go back to the same spot after reflecting integer multiples, would produce a bright spot at the opposite side against the diameter (from the top view) due to the propagation path inside the microsphere.

As shown in Figure R2b, the simulated field distribution also shows the pair of single mode lasing bright spots. The wide-field image (Figure R2a) was taken using a Prime 95B sCMOS camera through an 800 ± 20 nm bandpass filter, which contains multiple modes ($\text{TM}_{26,1}$, $\text{TE}_{26,1}$, $\text{TM}_{25,1}$ etc), thus showing a whole bright spot, instead of the striped spots. We have added more details in the revised manuscript, see lines 126-129, Page 5:

"As shown in Figure 2e, by focusing the 980 nm pump laser at the edge of the micro cavity, intense upconversion emissions are observed at the excitation spot and the edge of its opposite end due to the propagation path inside the microsphere, which suggests the onset of whispering gallery modes emissions."

Figure R2 **a**, Wide-field image of a lasing micro cavity pumped by the laser excitation of a diffraction-limited spot on the side of the microsphere (imaged using an Prime 95B sCMOS camera through a 800 ± 20 nm bandpass filter). **b**, Simulated 2D electric field distribution of a single mode at 800 nm in the XY plane (with a dipole on the edge).

We also calculated the order of the WGMs around 800 nm to get exact modes involved in the wide field image according to Mie Scattering, as shown in Figure R3.

Figure R3 Numerical simulation of resonance spectrum and experimental emission spectrum.

(4)-The discussion related to Figure 2b would benefit from the inclusion of the ratio between bands and how it depends on the Tm concentration.

Response: Thanks for this suggestion. We have expanded the discussion related to Figure 2b by adding the following sentences (lines 106-110, Page 4):

“The intensity at 802 nm ($^3H_4 \rightarrow ^3H_6$) transition, the intensity ratios of 802 nm/473 nm and 802 nm/645 nm reach their peak values at 2 mol. % Tm^{3+} due to the CR induced population enrichment at the excitation power density of $100 W/cm^2$. Further increase in the Tm^{3+} concentration, e.g. 4 mol%

and 8 mol.%, depopulates the 3H_4 level with decreased intensity at 802 nm due to the excessive CR induced energy loss and the possible back energy transfer from Tm^{3+} to Yb^{3+} .”

(5)- In the laser curves included in the SI the presence of a laser threshold is not so evident as in the data authors included in the main text. Why is this? How authors can be sure that what they are observing is laser emission instead of a spectral modulation of the spontaneous emission due to the presence of WGMs? Can authors measure the lifetime as a function of pump power to check for the stimulated emission induced lifetime shortening?

Response: Thanks for this suggestion.

We have plotted the laser curves in the log-log plot as suggested by reviewer #1. And we have enriched the testing points to get an evident inflection point, see Figure S13.

As suggested by this reviewer, we measured the power-dependent lifetime data, as shown in Figure R4. The lifetime shortens from 291 μ s to 158 μ s as the pump power increasing from 0.1 kW/cm^2 to 100 kW/cm^2 with a leap at the threshold. Besides, we also calculated the Purcell factor ($F_p = \frac{3}{4\pi^2} \left(\frac{\lambda}{n}\right)^3 \frac{Q}{V_{eff}}$, in which λ is lasing peak wavelength, n is refractive index, Q is quality factor and V_{eff} is mode volume) of the selected emission mode. And the simulated F_p of TE mode lasing at 800 nm is ~ 3.6 . This indicates that the lifetime could have been reduced about 3.6 fold, theoretically from 291 μ s to 80 μ s. Note that the decay curves were measured by using a 800 ± 20 nm bandpass filter, in which the broad signal collection window covered the multiple lasing peaks with the spontaneous emission background. This leads to the discrepancy between the experimental and theoretical values.

Figure S13 The systematic characterizations of the lasing threshold for the cavities coated with UCNP at Tm^{3+} doping of **a-e**, 0.2%, 0.5%, 1%, 4%, 8%.

Figure R4 Power-dependent lifetime decay curves of ³H₄ energy level related emissions through a 800±20 nm bandpass filter.

Reviewer #3 (Remarks to the Author):

In their manuscript Zhou et al. report on lasing emissions from single NaYF₄:Yb,Tm nanocrystals as part of a microcavity. By optimizing the Tm³⁺ concentration, they achieved upconverted laser emission at room temperature upon cw excitation with only 150 W/cm². This low lasing threshold is quite remarkable as such but also because it was achieved without reducing surface quenching of the emission by coating the nanocrystals with an inert shell. Their work significantly expands the potential of using upconverting nanocrystals for microscale lasers. The nanomaterials have been well characterized, all conclusions are supported by the data and the figures are of appropriate quality. I therefore support publication of the manuscript in Nature Communications without any modifications.

Response: We thank this reviewer for his/her very positive recommendation of our work.

- 1 Zou, W. Q., Visser, C., Maduro, J. A., Pshenichnikov, M. S. & Hummelen, J. C. Broadband dye-sensitized upconversion of near-infrared light. *Nat Photonics* **6**, 560-564 (2012).
- 2 Carnall, W., Goodman, G., Rajnak, K. & Rana, R. A systematic analysis of the spectra of the lanthanides doped into single crystal LaF₃. *The Journal of Chemical Physics* **90**, 3443-3457 (1989).
- 3 Dussardier, B., Blanc, W. & Peterka, P. Tailoring of the Local Environment of Active Ions in Rare-Earth- and Transition-Metal-Doped Optical Fibres, and Potential Applications. (InTech, 2012).
- 4 Foreman, M. R., Swaim, J. D. & Vollmer, F. Whispering gallery mode sensors. *Adv Opt Photonics* **7**, 168-240 (2015).

REVIEWERS' COMMENTS

Reviewer #1 (Remarks to the Author):

In the reply, the authors have addressed satisfactorily all my comments and concerns; they made proper changes to both main text and supplementary material. I believe the manuscript can be published in Nature Communications in its current form.

Reviewer #2 (Remarks to the Author):

I think authors have considered seriously all the points raised by the reviewers including me. Nevertheless I have some doubts about the way authors reply this point:

(3)- Why Figure 2e evidences the presence of WGMs? The appearance of an opposite bright spot would occur only for certain WGMs with a wavelength equal to the diameter of microsphere. Is this the case? which is the order of the WGMs in this case (it can be estimated from emission spectra).

Response: As we excite the gain medium on the edge of this WGM cavity with a diffraction-limited spot, the light, which can go back to the same spot after reflecting integer multiples, would produce a bright spot at the opposite side against the diameter (from the top view) due to the propagation path inside the microsphere.

As shown in Figure R2b, the simulated field distribution also shows the pair of single mode lasing bright spots. The wide-field image (Figure R2a) was taken using a Prime 95B sCMOS camera through an 800 ± 20 nm bandpass filter, which contains multiple modes (TM_{26,1}, TE_{26,1}, TM_{25,1} etc), thus showing a whole bright spot, instead of the striped spots. We have added more details in the revised manuscript, see lines 126-129, Page 5:

"As shown in Figure 2e, by focusing the 980 nm pump laser at the edge of the micro cavity, intense upconversion emissions are observed at the excitation spot and the edge of its opposite end due to the propagation path inside the microsphere, which suggests the onset of whispering gallery modes emissions."

I can not see the point why the use of band-pass filters only allows to see one particular WG mode that is the one leading to a bright spot opposite to the pump spot. It is clear that WG Modes are present in their experiments (peaks in the emission spectra) but I am afraid that authors try to use this image as a proof of the WGM existence. I am afraid this could not be correct. As it is not essential in their manuscript I would suggest authors to remove this figure unless they can give a much more convincing explanation

Response Letter

Our point-by-point responses (in blue) to the reviewers' comments (in black) are provided below.

Reviewer #1 (Remarks to the Author):

In the reply, the authors have addressed satisfactorily all my comments and concerns; they made proper changes to both main text and supplementary material. I believe the manuscript can be published in Nature Communications in its current form.

Response: We thank this reviewer for his/her overall positive comments.

Reviewer #2 (Remarks to the Author):

I think authors have considered seriously all the points raised by the reviewers including me.

Nevertheless I have some doubts about the way authors reply this point:

(3)- Why Figure 2e evidences the presence of WGMs? The appearance of an opposite bright spot would occur only for certain WGMs with a wavelength equal to the diameter of microsphere. Is this the case? which is the order of the WGMs in this case (it can be estimated from emission spectra).
Response: As we excite the gain medium on the edge of this WGM cavity with a diffraction-limited spot, the light, which can go back to the same spot after reflecting integer multiples, would produce a bright spot at the opposite side against the diameter (from the top view) due to the propagation path inside the microsphere.

As shown in Figure R2b, the simulated field distribution also shows the pair of single mode lasing bright spots. The wide-field image (Figure R2a) was taken using a Prime 95B sCMOS camera through an 800 ± 20 nm bandpass filter, which contains multiple modes ($TM_{26,1}$, $TE_{26,1}$, $TM_{25,1}$ etc), thus showing a whole bright spot, instead of the striped spots. We have added more details in the revised manuscript, see lines 126-129, Page 5:

“As shown in Figure 2e, by focusing the 980 nm pump laser at the edge of the micro cavity, intense upconversion emissions are observed at the excitation spot and the edge of its opposite end due to the propagation path inside the microsphere, which suggests the onset of whispering gallery modes emissions.”

I can not see the point why the use of band-pass filters only allows to see one particular WG mode that is the one leading to a bright spot opposite to the pump spot. It is clear that WG Modes are present in their experiments (peaks in the emission spectra) but I am afraid that authors try to use this image as a proof of the WGM existence. I am afraid this could not be correct. As it is not essential in their manuscript I would suggest authors to remove this figure unless they can give a much more convincing explanation.

Response: Thanks this reviewer for bringing this issue for further discussion. Due to the propagation path of lasing modes inside the microspheric cavity, there will be a brighter spot at the opposite end when we focus the pump laser at the edge of the micro cavity. However, this might lead to unnecessary misunderstanding or disputation. As suggested by the reviewer, we have removed this sub-figure in Figure 2.